# GOPlan: Goal-conditioned Offline Reinforcement Learning by Planning with Learned Models

**Mianchu Wang**[*]                                                                    *Mianchu.Wang@warwick.ac.uk*
*University of Warwick*

**Rui Yang**[*]                                                                              *ryangam@connect.ust.hk*
*Hong Kong University of Science and Technology*

**Xi Chen**                                                                                  *pcchenxi@tsinghua.edu.cn*
*Tsinghua University*

**Hao Sun**                                                                                        *hs789@cam.ac.uk*
*University of Cambridge*

**Meng Fang**[†]                                                                          *Meng.Fang@liverpool.ac.uk*
*University of Liverpool*

**Giovanni Montana**[†]                                                             *g.montana@warwick.ac.uk*
*University of Warwick*

**Reviewed on OpenReview:** *https://openreview.net/forum?id=zOKAmm8R9B*

## Abstract

Offline Goal-Conditioned RL (GCRL) offers a feasible paradigm for learning general-purpose policies from diverse and multi-task offline datasets. Despite notable recent progress, the predominant offline GCRL methods, mainly model-free, face constraints in handling limited data and generalizing to unseen goals. In this work, we propose *Goal-conditioned Offline Planning* (GOPlan), a novel model-based framework that contains two key phases: (1) pretraining a prior policy capable of capturing multi-modal action distribution within the multi-goal dataset; (2) employing the *reanalysis* method with planning to generate imagined trajectories for funetuning policies. Specifically, we base the prior policy on an advantage-weighted conditioned generative adversarial network, which facilitates distinct mode separation, mitigating the pitfalls of out-of-distribution (OOD) actions. For further policy optimization, the *reanalysis* method generates high-quality imaginary data by planning with learned models for both intra-trajectory and inter-trajectory goals. With thorough experimental evaluations, we demonstrate that GOPlan achieves state-of-the-art performance on various offline multi-goal navigation and manipulation tasks. Moreover, our results highlight the superior ability of GOPlan to handle small data budgets and generalize to OOD goals.

## 1 Introduction

Offline reinforcement learning (RL) (Fujimoto et al., 2019; Kumar et al., 2020; Levine et al., 2020; Janner et al., 2021) enables learning policies from offline dataset without online interactions with the environment, offering an efficient and safe approach for real-world applications, e.g., robotics, autonomous driving, and healthcare (Levine et al., 2020). Developing a general-purpose policy capable of multiple skills is particularly appealing for RL. Offline goal-conditioned RL (GCRL) (Chebotar et al., 2021; Yang et al., 2022b) offers such a way to learn multiple skills from diverse and multi-goal datasets. So far, prior works in this

---

[*]Equal contribution. [†]Corresponding authors.

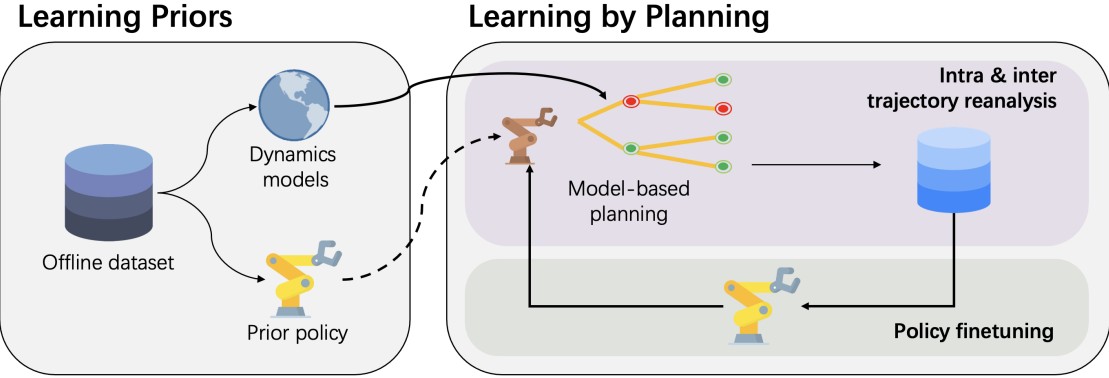

Figure 1: The two-stage framework of GOPlan: pretraining a prior policy and a group of dynamics models, and finetuning policy with imagined trajectories generated by the reanalysis method.

direction (Chebotar et al., 2021; Yang et al., 2022b; Ma et al., 2022; Mezghani et al., 2022) mainly focus on avoiding out-of-distribution (OOD) actions and learning to reach all in-dataset goals. Despite recent progress, efficiently mastering diverse skills with limited data and achieving OOD goal generalization remain significant challenging.

Model-based RL (Schrittwieser et al., 2020; Hafner et al., 2020; Moerland et al., 2023) is a natural choice to overcome these two challenges for offline GCRL, as it enables efficient learning from limited data (Deisenroth & Rasmussen, 2011; Argenson & Dulac-Arnold, 2021; Schrittwieser et al., 2021) and enjoys better generalization (van Steenkiste et al., 2019; Yu et al., 2020; Lee et al., 2020). The advantages of model-based RL in addressing the aforementioned challenges emanate from its ability to utilize more direct supervision information rather than relying solely on scalar rewards. Additionally, world models used in model-based RL are trained using supervised learning, which is more stable and reliable than bootstrapping (Yu et al., 2020). A recent notable model-based method, *reanalyse* (Schrittwieser et al., 2020; 2021), employs model-based value estimation and planning to generate improved value and policy target for given states, showing advantageous performance for both online and offline RL and holding the potential for handling the limited data and OOD generalization challenges. However, it should be noted that reanalysis is primarily designed for single-task RL and cannot be directly applied for offline GCRL. This is in part due to the challenges presented by multi-goal datasets, which contain trajectories collected for heterogeneous goals. For effective long-term planning in offline GCRL, accurately modeling action modes in these datasets while avoiding OOD actions is crucial. In addition, the question of what goal to plan for, must be addressed in reanalysis for offline GCRL, which is not a concern for single-task RL.

To this end, we introduce *Goal-conditioned Offline Planning* (GOPlan), a novel model-based algorithm designed for offline GCRL. A diagram of GOPlan is shown in Figure 1. GOPlan consists of two main stages, a pretraining stage and a reanalysis stage. During the pretraining stage, GOPlan trains a policy using advantage-weighted Conditioned Generative Adversarial Network (CGAN) to capture the multi-modal action distribution in the heterogeneous multi-goals dataset. The pretrained policy exhibits notable mode separation to avoid OOD actions and is improved towards high-reward actions, making it suitable for offline planning. A group of dynamics models is also learned during this stage and will be used for planning and uncertainty quantification. During the reanalysis stage, GOPlan finetunes the policy with imagined trajectories for further policy optimization. Specifically, GOPlan generates a reanalysis buffer by planning using the policy and learned models for both intra-trajectory and inter-trajectory goals. Given a state in a trajectory, intra-trajectory goals are along the same trajectory as the state, and inter-trajectory goals lie in different trajectories. We quantify the uncertainty of the planned trajectories based on the disagreement of the models (Pathak et al., 2019; Yu et al., 2020) to avoid going excessively outside the dynamics model's support. The imagined data with small uncertainty can be high-quality demonstrations that can enhance the agent's ability to achieve both in-dataset and out-of-dataset goals. By iteratively planning to generate better data and finetune the policy with advantage-weighted CGAN, the reanalysis stage significantly improves the

policy's performance while also reducing the requirement for a large offline dataset. Furthermore, GOPlan can leverage the generalization of dynamics models to plan for OOD goals, which is a particularly promising feature for offline GCRL.

In the experiments, we evaluate GOPlan across multiple multi-goal navigation and manipulation tasks, and demonstrate that GOPlan outperforms prior state-of-the-art (SOTA) offline GCRL algorithms. Moreover, we extend the evaluation to the small data budget and the OOD goal settings, in which GOPlan exhibits superior performance compared to other algorithms by a substantial margin. To summarize, our main contributions are three folds:

1. We propose GOPlan, a novel model-based offline GCRL algorithm that can address challenging settings with limited data and OOD goals.

2. GOPlan consists of two stages, pretraining a prior policy via advantage-weighted CGAN and fine-tuning the policy with high-quality imagined trajectories generated by planning.

3. Experiments validate GOPlan's efficacy in benchmarks as well as two challenging settings.

## 2 Related Work

**Offline RL.** Addressing OOD actions for offline RL is essential to ensure safe policies during deployment. Several works propose to avoid OOD actions by enforcing constraints on the policy (Wang et al., 2018; Wu et al., 2019; Nair et al., 2021) or penalizing the Q values of OOD state-action pairs (Kumar et al., 2020; An et al., 2021; Bai et al., 2022; Yang et al., 2022a; Sun et al., 2022). However, these methods generally use a uni-modal Gaussian policy, which cannot capture the multi-modal action distribution of the heterogeneous offline dataset. To address the multi-modality issue, PLAS (Zhou et al., 2020) decodes variables sampled from a variational auto-encoder (VAE) latent space, while LAPO (Chen et al., 2022) leverages advantage-weighted latent space to further optimize the policy. A recent work (Yang et al., 2022c) demonstrates GAN's ability to capture multiple action modes. Based on GAN, DASCO (Vuong et al., 2022) proposes a dual-generator algorithm to maximize the expected return. Distinctively, we propose to capture the multi-modal distribution in multi-goal dataset via advantage-weighted CGAN.

**Offline GCRL.** In offline GCRL, agents learn goal-conditioned policies from a static dataset. One direction for offline GCRL is through goal-conditioned supervised learning (GCSL), which directly performs imitation learning on the relabelled data (Ghosh et al., 2021; Emmons et al., 2022). When the data is suboptimal or noisy, weighted GCSL (Yang et al., 2022b; Wang et al., 2024; Yang et al., 2023) with multiple weighting criteria is a more powerful solution. Other directions include optimizing the state occupancy on the targeted goal distribution (Ma et al., 2022) and contrastive RL (Eysenbach et al., 2022). Additionally, Mezghani et al. (2022) design a self-supervised dense reward learning method to solve offline GCRL with long-term planning. Different from prior works, our work alleviates the multi-modality problem raised from the multi-goal dataset via advantage-weighted CGAN, and efficiently utilizes dynamics models for reanalysis and policy improvement.

**Model-based RL.** The ability to reach any in-dataset goals requires an agent to grasp the invariance underlying different tasks. The invariance can be represented by the dynamics of the environment that can facilitate RL (Schrittwieser et al., 2020; Hafner et al., 2020; Nagabandi et al., 2020; Hansen et al., 2022; Rigter et al., 2022) and GCRL (Charlesworth & Montana, 2020; Yang et al., 2021). In the offline setting, an ensemble of dynamics models is used to construct a pessimistic MDP (Kidambi et al., 2020; Yu et al., 2020); Reanalysis (Schrittwieser et al., 2021) uses Monte-Carlo tree search (Coulom, 2007) with learned model to generate new training targets for a given state. These dynamics models could be deterministic models (Yang et al., 2021), Gaussian models (Nagabandi et al., 2020), or recurrent models with latent space (Hafner et al., 2019; 2020). GOPlan chooses the deterministic models due to its simplicity, and with the recurrent models it can be extended to high-dimensional state space. Imagined trajectories during offline planning may contain high uncertainty, resulting in inferior performance. MOPP (Zhan et al., 2022) measures the uncertainty by the disagreement of the dynamics models and prunes uncertain trajectories. However, prior

model-based offline RL works are not designed for the multi-goal setting. Instead, we perform model-based planning for both intra-trajectory and inter-trajectory goals, iteratively finetuning the GAN-based policy with high-quality and low-uncertainty imaginary data.

## 3 Preliminaries

**Goal-conditioned RL.** Goal-conditioned RL is generally formulated as Goal-Conditioned Markov Decision Processes (GCMDP), denoted by a tuple $< \mathcal{S}, \mathcal{A}, \mathcal{G}, P, \gamma, r >$ where $\mathcal{S}$, $\mathcal{A}$, and $\mathcal{G}$ are the state, action and goal spaces, respectively. $P(s_{t+1} \mid s_t, a_t)$ is the transition dynamics, $r$ is the reward function, and $\gamma$ is the discount factor. An agent learns a goal-conditioned policy $\pi : \mathcal{S} \times \mathcal{G} \to \mathcal{A}$ to maximize the expected discounted cumulative return:

$$J(\pi) = \mathbb{E}_{\substack{g \sim P_g, s_0 \sim P_0(s_0), \\ a_t \sim \pi(\cdot|s_t,g), \\ s_{t+1} \sim P(\cdot|s_t,a_t)}} \left[ \sum_{t=0}^{T} \gamma^t r(s_t, a_t, g) \right], \tag{1}$$

where $P_g$ and $P_0(s_0)$ are the distribution of the goals and initial states, and $T$ corresponds to the length of an episode. The expected value of a state-goal pair is defined as: $V^\pi(s, g) = \mathbb{E}_\pi \left[ \sum_{t=0}^{T} \gamma^t r_t \mid s_0 = s \right]$. A sparse reward is non-zero only when the goal is reached, i.e., $r(s, a, g) = \mathbb{1}[\|\phi(s) - g\|_2^2 \leq \epsilon]$, where $\phi : \mathcal{S} \to \mathcal{G}$ is the state-to-goal mapping (Andrychowicz et al., 2017) and $\epsilon$ is a threshold. This sparse reward function is accessible to the agent during the learning process. For offline GCRL, the agent can only learn from an offline dataset $\mathcal{B}$ without interacting with the environment.

**Reanalysis.** MuZero Reanalyse (Schrittwieser et al., 2020; 2021) introduces reanalysis to perform iterative model-based value and policy improvement for states in the dataset, resulting in an ongoing cycle of refinement through updated predictions and improved generated training data. Specifically, MuZero Reanalyse updates its parameters $\theta$ for $K$-step model rollouts via supervised learning:

$$l_t(\theta) = \sum_{k=0}^{K} l^p(\pi_{t+k}, p_t^k) + \sum_{k=0}^{K} l^v(z_{t+k}, v_t^k) + \sum_{k=0}^{K} l^r(u_{t+k}, r_t^k), \tag{2}$$

where $p_t^k, v_t^k, r_t^k$ are action, value, and reward predictions generated by the model $\theta$. Their training targets $\pi_{t+k}, z_{t+k}$ are generated via search tree, and $u_{t+k}$ is the true reward. In continuous control settings, the loss functions $l^p(\cdot, \cdot)$, $l^v(\cdot, \cdot)$, $l^r(\cdot, \cdot)$ are mean square error (MSE) between the inputs.

## 4 Methodology

This section introduces the two-stage GOPlan algorithm, specially designed for offline GCRL. During the pretraining stage, we implement an advantage-weighted CGAN to establish a proficient prior policy for capturing multi-modal action distribution in offline datasets, which is suitable for subsequent model-based planning. In the reanalysis stage, we enhance the performance of the agent by enabling planning with learned models and multiple goals, resulting in a significant policy improvement. The overall framework is depicted in Figure 1.

### 4.1 Learning Priors from Offline Data

**Learning Prior Policy.** The initial step involves learning a policy that can generate in-distribution and high-reward actions from multi-goal offline data. Due to the nature of collecting data for multiple heterogeneous goals, these datasets can be highly multi-modal (Lynch et al., 2020; Yang et al., 2022b), meaning that a state can have multiple valid action labels. The potential conflict between these actions poses a significant learning challenge. Unlike prior works using Gaussian (Yang et al., 2022b; Ma et al., 2022), conditional Variational Auto-encoder (CVAE) (Zhou et al., 2020; Chen et al., 2022) and Conditioned Generative Adversarial Network (CGAN) (Yang et al., 2022c), we employ Weighted CGAN as the prior policy. In Figure

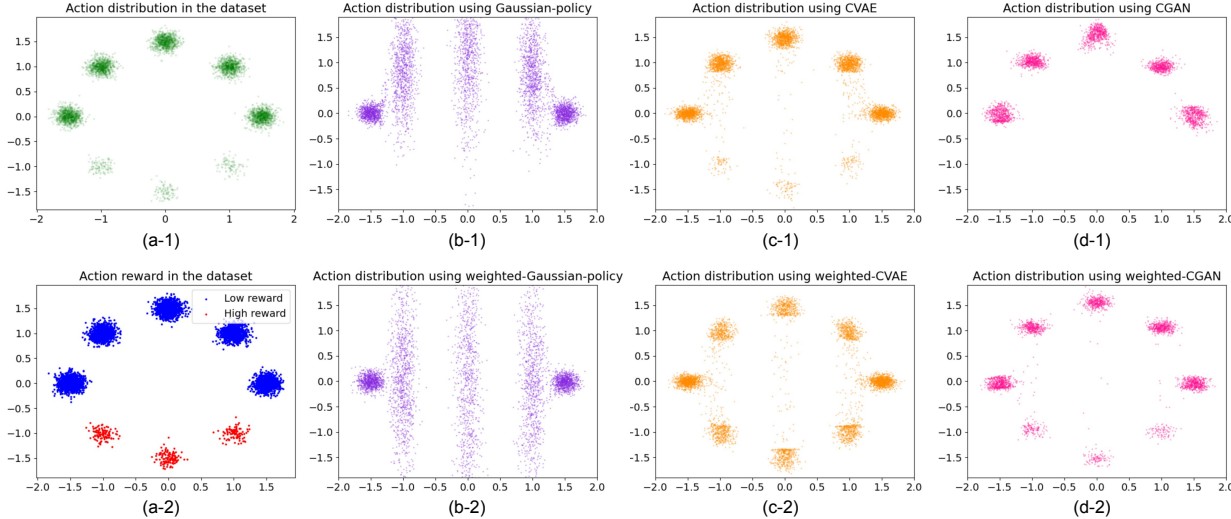

Figure 2: An example about modeling the multi-modal behavior policy while maximizing average rewards. The $x$-axis represents the state, and the $y$-axis represents the multi-modal action. (a-1) shows the action distribution of the offline dataset. (a-2) shows the corresponding reward distribution. (b-1) and (b-2) illustrate the action distributions generated by Gaussian and Weighted Gaussian. (c-1) and (c-2) illustrate action distributions from CVAE and Weighted CVAE. (d-1) and (d-2) illustrate action distributions from CGAN and Weighted CGAN.

2, we compare six different models : Gaussian, Weighted Gaussian, CVAE, Weighted CVAE, CGAN, and Weighted CGAN on a multi-modal dataset with imbalanced rewards, where high rewards have less frequency, as illustrated in Figure 2(a-2). The weighting scheme for Weight CVAE and Weighted CGAN is based on advantage re-weighting (Yang et al., 2022b) and we directly use rewards as weights in this example.

In Figure 2, Weighted CGAN outperforms other models by exhibiting a more distinct mode separation. As a result, the policy generates fewer OOD actions by reducing the number of interpolations between modes. In contrast, other models all suffer from interpolating between modes. Even though VAE models perform better than Gaussian models, they are still prone to interpolation due to the regularization of the Euclidean norm on the Jacobian of the VAE decoder (Salmona et al., 2022). Furthermore, without employing advantage-weighting, both the CVAE and CGAN models mainly capture the denser regions of the action distribution, but fail to consider their importance relative to the rewards associated with each mode.

Based on the empirical results, the utilization of the advantage-weighted CGAN model for modeling the prior policy from multi-modal offline data demonstrates notable advantages for offline GCRL. In this framework, the discriminator is responsible for distinguishing high-quality actions in the offline dataset from those generated by the policy, while the generative policy is designed to generate actions that outsmart the discriminator in an adversarial process. This mechanism encourages the policy to produce actions that closely resemble high-quality actions from the offline dataset. In Section 4.3, we will elaborate on the specific definition of the advantage-weighted CGAN model.

**Learning Dynamics Models.** We also train a group of $N$ dynamic models $\{M_{\psi_i}\}_{i=1}^{N}$ from the offline data to predict the residual between current states and next states. Each model $M_{\psi_i}$ minimizes the following loss:

$$\mathcal{L}(\psi_i) = \mathbb{E}_{(s_t, a_t, s_{t+1}) \sim \mathcal{B}} \left[ ||M_{\psi_i}(s_t, a_t) - (s_{t+1} - s_t)||_2^2 \right]. \tag{3}$$

Then the predicted next state is $\overline{s}_{t+1} = s_t + M_{\psi_i}(s_t, a_t)$. As the dynamics models are trained through supervised learning, making them more stable and better equipped with generalization abilities compared to bootstrapping (Yu et al., 2020). In the subsequent section, we will illustrate how the learned models can be utilized to enhance the training efficiency and generalization ability of offline GCRL.

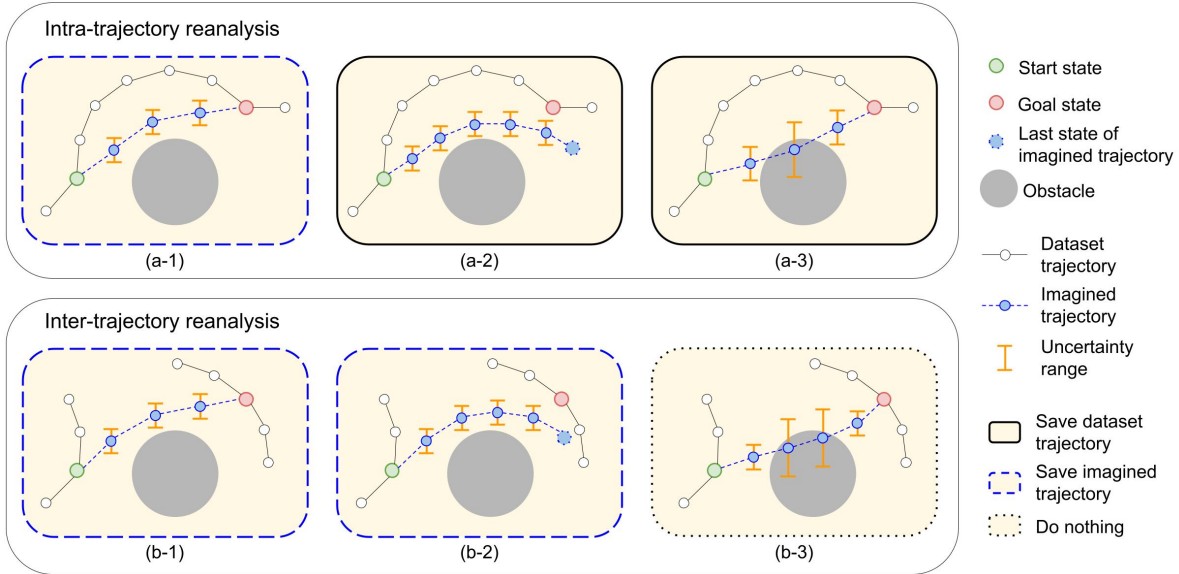

Figure 3: Illustration of intra-trajectory and inter-trajectory reanalysis. There are six scenarios: (a-1) the imagined trajectory is valid and better than the original trajectory; (a-2) the imagined trajectory fails to reach the goal within the same number of steps as the original trajectory; (b-1) a valid imagined trajectory connects the state to an inter-trajectory goal; (b-2) a valid imagined trajectory that does not achieve the desired goal; (a-3) (b-3) invalid imagined trajectories with large uncertainty.

## 4.2 Learning by Planning with Learned Models

Based on the learned prior policy and dynamics models, we reanalyse and finetune the policy in this learning procedure. As the dynamics information encapsulated in the offline dataset remains invariant across different goals, we utilize this property to equip our policy with the capacity to achieve a broad spectrum of goals by reanalysing the current policy for both intra-trajectory and inter-trajectory goals. The reanalysis procedure is shown in Figure 3, where we apply model-based planning method to generate trajectories for selected goals and save potential trajectories that can help improve the policy and have small uncertainty. By integrating this imagined data to update the policy, we repeat the aforementioned process to iteratively improve the policy.

**Model-based Planning.** We first introduce our planning method that serves intra-trajectory and inter-trajectory reanalysis. Our planner is similar to prior model-predictive approaches (Nagabandi et al., 2020; Argenson & Dulac-Arnold, 2021; Charlesworth & Montana, 2020; Schrittwieser et al., 2021), where the planner employs a model to simulate multiple imaginary trajectories, assigns scores to the initial actions, and generates actions based on these scores. Following prior work (Charlesworth & Montana, 2020), given the current state $s_t$ and the selected goal $g$, our planner firstly samples $C$ initial actions $\{a_t^c\}_{c=1}^C$ from the prior generative policy $\pi$ and predicts $C$ next states $\{\overline{s}_{t+1}^c\}_{c=1}^C$ based on a randomly selected dynamics model $M_{\psi_i}$. To score each initial action, the planner duplicates every next state $H$ times and generates $H$ imagined trajectories of $K$ steps with the policy and a randomly chosen dynamics model at each step. For each initial action $a_t^c$, the planner averages and normalizes all cumulative returns commencing from that initial action as $R(a_t^c)$, where the reward for every generated step is computed by the sparse reward function. Finally, the planner's output action for $s_t, g$ is $a_t = \frac{\sum_{c=1}^C e^{\kappa R(a_t^c)} a_t^c}{\sum_{c=1}^C e^{\kappa}}$, where $\kappa$ is a hyper-parameter tuning the exponentially weights for actions. It is worth noting that advantage-weighted CGAN effectively provides diverse in-dataset actions with higher rewards, enabling efficient and safe action planning for offline GCRL. The planning algorithm is outlined in Algorithm 2 provided in Appendix A.

**Intra-trajectory and Inter-trajectory Reanalysis.** Offline goal-conditioned planning involves the challenges of choosing planning goals and selecting appropriate imaginary data for policy training. In this paper, we present a novel approach that provides solutions to these challenges by incorporating intra-trajectory and inter-trajectory reanalysis, as illustrated in Figure 3. Specifically, our approach includes designed mechanisms for assigning goals and selecting generated trajectory to be stored in a reanalysis buffer $\mathcal{B}_{re}$ for subsequent policy improvement.

Since the primary objective of offline GCRL is to reach all in-dataset goals, utilizing all the states in the dataset as goals appears to be a natural choice. These goals can be classified into two categories: ***intra-trajectory goals***, which are along the same trajectory as the starting state, and ***inter-trajectory goals***, which lie in different trajectories. The division criterion is determined by the existence of a path between the state and the goal. Intra-trajectory goals refer to those that already have a path that links the state to them and hence, they can be reached with high probability because the successful demonstration has been provided. In contrast, inter-trajectory goals may not have a valid path to be reached from the given state. Therefore, different criterion is needed for the two situations.

The main idea of the two types of reanalysis is to store improved trajectories than the dataset and remove invalid trajectories with large uncertainty. The criterion of removing unrealistic imagined trajectories with large uncertainty is shared by both intra-trajectory and inter-trajectory reanalysis, because imitating these trajectories can lead the policy excessively outside the support of the dynamics model. To achieve this, we measure the the uncertainty of a trajectory by accessing the disagreement of a group of learned dynamics models. Specifically, given a trajectory $\tau = \{s_0, a_0, s_1, a_1, ..., s_H\}$, the uncertainty is defined as the maximal disagreement on the transition within the trajectory:

$$U(\tau) = \max_{0 \le k \le H} \frac{1}{N} \sum_{i=1}^{N} ||M_{\psi_i}(s_k, a_k) - \bar{s}_{k+1}||_2^2, \tag{4}$$

where $M_{\psi_i}$ is the $i$-th model and $\bar{s}_{k+1}$ is the mean prediction over $N$ models: $\bar{s}_{k+1} = \frac{1}{N} \sum_{i=1}^{N} M_{\psi_i}(s_k, a_k)$.

***Intra-trajectory reanalysis.*** randomly samples a trajectory $\tau$ and two states $s_t, s_{t+k} \in \tau$, where $k > 0$. After that, the model-based planning approach mentioned earlier is then employed to generate an action at each step from $s_t$ with the aim of reaching $\phi(s_{t+k})$, transiting according to the mean prediction of $M_{\psi_i}, i \in [1, N]$. There are mainly three cases: (a-1) if the planner reaches $\phi(s_{t+k})$ with fewer than $k$ steps and the uncertainty of the imagined trajectory is less than a threshold value $u$, the generated virtual trajectory $\bar{\tau}$ is stored into a reanalysis buffer $\mathcal{B}_{re}$. In cases where the (a-2) policy fails to reach $\phi(s_{t+k})$ within $k$ steps or (a-3) the generated virtual trajectory has larger uncertainty than $u$, the original trajectory $\tau_{t:t+k}$ is included in $\mathcal{B}_{re}$ because we can enhance learning the original successful demonstrations during fine-tuning.

***Inter-trajectory reanalysis.*** involves randomly selecting a state $s_t$ in a trajectory $\tau_1$ and another state $s_g$ in a different trajectory $\tau_2$, followed by model-based planning from $s_t$ to reach $\phi(s_g)$. There are also three cases: (b-1) if the planner successfully reaches $\phi(s_g)$ with an uncertainty less than $u$, we include the generated virtual trajectory $\bar{\tau}$ into the reanalysis buffer $\mathcal{B}_{re}$. (b-2) Moreover, in situations where the policy fails to attain the intended goal state $\phi(s_g)$ but the associated trajectory exhibits low levels of uncertainty, the virtually achieved goals along the generate trajectory are labeled as the intended goal for the trajectory. This approach enables the utilization of failed virtual trajectories to enhance the diversity of the reanalysis buffer $\mathcal{B}_{re}$. (b-3) Additionally, if the imagined trajectory has high uncertainty, we do not save such invalid trajectory.

## 4.3 Policy Optimization

In GOPlan, the policy is updated during both the pretraining stage and the finetuning stage. We train the policy $\pi$ via advantage-weighted CGAN, which can be formulated as the following objective:

$$\max_D \min_\pi \mathbb{E}_{(s_t, a_t, g) \sim \mathcal{B}} \left[ w(s_t, a_t, g) \log D(s_t, a_t, g) \right]$$
$$+ \mathbb{E}_{(s_t, g) \sim \mathcal{B}, a' \sim \pi} \left[ \log(1 - D(s_t, a', g)) \right] \tag{5}$$

Here, $D$ is the discriminator weighted by $w$. $w(s, a, g) = \exp(A(s, a, g) + N(s, g))$ is the exponential advantage weight, where $N(s, g)$ serves as a normalizing factor to ensure that $\sum_{a \in \mathcal{A}} w(s, a, g) \pi_b(a \mid s, g) = 1$. $\pi_b$ is the behavior policy underlying the relabeled offline dataset $\mathcal{B}$. The advantage function can be estimated by a learned value function $V_{\theta_v}$: $A(s_t, a_t, g) = r(s_t, a_t, g) + \gamma V_{\theta_v}(s_{t+1}, g) - V_{\theta_v}(s_t, g)$. The value function $V_{\theta_v}(s_t, g)$ is updated to minimize the TD loss:

$$\mathcal{L}(\theta_v) = \mathbb{E}_{(s_t, r_t, s_{t+1}, g) \sim \mathcal{B}} \left[ (V_{\theta_v}(s_t, g) - y_t)^2 \right], \tag{6}$$

where the target $y_t = r_t + \gamma V_{\theta_v^-}(s_{t+1}, g)$ and the target network $V_{\theta_v^-}$ is slowly updated to improve training stability.

To optimize the minmax objective in Eq.equation 5, we train a discriminator $D$ and a generative policy $\pi$ separately. The discriminator $D$, with parameter $\theta_d$, learns to minimize the following loss function:

$$\mathcal{L}(\theta_d) = - \mathbb{E}_{(s_t, a_t, g) \sim \mathcal{B}} \left[ w(s_t, a_t, g) \log D_{\theta_d}(s_t, a_t, g) \right]$$
$$- \mathbb{E}_{\substack{(s_t, g) \sim \mathcal{B}, z \sim P(z), \\ a' \sim \pi_{\theta_\pi}(s_t, g, z)}} \left[ \log(1 - D_{\theta_d}(s_t, a', g)) \right], \tag{7}$$

where $P(z)$ constitutes random noise that follows a diagonal Gaussian distribution. The discriminator's output is passed through a sigmoid function, thereby constraining the output to lie within $(0, 1)$. To deceive the discriminator, the policy network $\pi$, with parameter $\theta_\pi$, is trained to minimize the following loss function:

$$\mathcal{L}(\theta_\pi) = \mathbb{E}_{\substack{(s_t, a_t, g) \sim \mathcal{B}, z \sim P(z), \\ a' \sim \pi_{\theta_\pi}(s_t, g, z)}} \left[ \log(1 - D_{\theta_d}(s_t, a', g)) \right]. \tag{8}$$

Through this training process, the policy network is capable of producing actions that closely resemble high-quality actions in the dataset and can enjoy policy improvement similar to prior advantage-weighted works (Wang et al., 2018; Peng et al., 2019).

### 4.4 Overall Algorithm

In the pretraining stage, we train the dynamics models $M_{\psi_i}, i \in [1, N]$, the discriminator $D_{\theta_d}$, the value function $V_{\theta_v}$ and the policy $\pi_{\theta_\pi}$ until convergence. In the subsequent reanalysis stage, we generate new trajectories through intra-trajectory and inter-trajectory reanalysis, save them into the reanalysis buffer, and then employ the reanalysis buffer to finetune the value function, the discriminator, and the policy. The process of reanalysis and fine-tuning is repeated over $I$ iterations to improve the policy performance. Algorithm 1 outlines the details of the two main stages of the overall algorithm.

## 5 Experiments

We conduct a comprehensive evaluation of GOPlan across a collection of continuous control tasks (Plappert et al., 2018; Yang et al., 2022b) with multiple goals and sparse rewards. In addition, we also assess the efficacy of GOPlan in settings with limited data budgets and OOD goal generalization.

**Environments and Datasets.** To conduct benchmark experiments, we utilize offline datasets from (Yang et al., 2022b). The dataset contains $1 \times 10^5$ transitions for low-dimensional tasks and $2 \times 10^6$ for four high-dimensional tasks (FetchPush, FetchPick, FetchSlide and HandReach). Furthermore, to demonstrate the ability to handle small data budgets, we also integrate an additional group of small and extra small datasets, referred to with a suffix "-s" and "-es", containing only 10% and 1% of the number of transitions. To assess GOPlan's ability to generalize to OOD goals, we leverage four task groups from (Yang et al., 2023): FetchPush Left-Right, FetchPush Near-Far, FetchPick Left-Right, and FetchPick Low-High, each consisting of a dataset and multiple tasks. For instance, the dataset of FetchPush Left-Right contains trajectories where both the initial object and achieved goals are on the right side of the table. As such, the independent identically distributed (IID) task assesses agents handling object and goals on the right side (i.e., Right2Right), while the other tasks in the group assess OOD goals or starting positions, such as Right2Left, Left2Right, and Left2Left. For further information regarding the environments and datasets used in our evaluation, we refer readers to Appendix B.

---

**Algorithm 1** Goal-conditioned Offline Planning (GOPlan).

---

**Initialise:** $N$ dynamics models $\{\psi_i\}_{i=1}^N$, a discriminator $\theta_d$, a policy $\theta_\pi$, a goal-conditioned value function $\theta_v$; an offline dataset $\mathcal{B}$ and a reanalysis buffer $\mathcal{B}_{re}$; the state-to-goal mapping $\phi$.

```
 1: # Pre-train
 2: while not converges do
 3:    Update {ψ_i}_{i=1}^N using B.
 4:    Update θ_v using B.              ▷ Eq. 6
 5:    Update θ_d using B.              ▷ Eq. 7
 6:    Update θ_π using B.              ▷ Eq. 8
 7: end while
 8:
 9: # Finetune
10: for i = 1, ..., I do
11:    for j = 1, ..., I_intra do
12:       τ = Intra_traj()
13:       B_re = B_re ∪ τ
14:    end for
15:    for j = 1, ..., I_inter do
16:       τ = Inter_traj()
17:       B_re = B_re ∪ τ
18:    end for
19:    Finetune θ_v using B_re.         ▷ Eq. 6
20:    Finetune θ_d using B_re.         ▷ Eq. 7
21:    Finetune θ_π using B_re.         ▷ Eq. 8
22: end for
```

```
def Intra_traj():
 1: (s_t, s_{t+1}, ..., s_{t+K}, g) ~ B, ŝ_t = s_t
 2: for k = 0, ..., K do
 3:    a_{t+k} = Plan(ŝ_{t+k}, φ(s_{t+K}))
 4:    ŝ_{t+k+1} = M_{ψ_{i,i~{1,...,N}}}(ŝ_{t+k}, a_{t+k})
 5:    if U(ŝ_{t+k}, a_{t+k}) > u then
 6:       return {s_t, s_{t+1}, ..., s_{t+K}, g}.
 7:    end if
 8:    if ŝ_{t+k+1} achieves φ(s_{t+K}) then
 9:       return {s_t, ŝ_{t+1}, ..., ŝ_{t+k+1}, φ(s_{t+K})}.
10:    end if
11: end for
12: return {s_t, s_{t+1}, ..., s_{t+K}, g}.
def Inter_traj():
13: s_0 ~ B, s_g ~ B, ŝ_0 = s_0
14: for t = 0, ..., T do
15:    a_t = Plan(ŝ_t, φ(s_g))
16:    ŝ_{t+1} = M_{ψ_{i,i~{1,...,N}}}(ŝ_t, a_t)
17:    if U(ŝ_t, a_t) > u then
18:       return {∅}
19:    end if
20:    if ŝ_{t+1} achieves φ(s_g) then
21:       return {s_0, ŝ_1, ..., ŝ_{t+1}, φ(s_g)}
22:    end if
23: end for
24: return {s_0, ŝ_1, ..., ŝ_T, φ(ŝ_T)}
```

---

**Experimental Setup.** We compare GOPlan against SOTA offline GCRL algorithms, WGCSL (Yang et al., 2022b), contrastive RL (CRL) (Eysenbach et al., 2022), actionable models (AM) (Chebotar et al., 2021), GCSL (Ghosh et al., 2021), and modified offline RL methods, such as TD3-BC (g-TD3-BC) (Fujimoto & Gu, 2021), exponential advantage weighting (GEAW) (Wang et al., 2018), Trajectory Transformer (TT) (Janner et al., 2021), Decision Transformer (DT) (Chen et al., 2021). We denote a variant approach of GOPlan as **GOPlan2**, which employs testing-time model-based planning with candidate actions from the GOPlan policy. The testing-time planning method aligns with the model-based planning approach described in Section 4.2. For all experiments, we report the average returns with standard deviation across 5 different random seeds. Implementation details are provided in Appendix C.

## 5.1 Benchmarks and Small Dataset Results

Table 1 demonstrates the performance of GOPlan and baselines in the benchmark tasks. As shown in the table, GOPlan shows improved performance over current SOTA model-free algorithms, achieving the highest average return on 8 out 10 tasks. Notably, unlike prior online model-based GCRL approaches (Charlesworth & Montana, 2020; Yang et al., 2021) that fail to handle high dimensional tasks like HandReach, GOPlan works well for these tasks. This perhaps can be attributed to the fact that GOPlan incorporates both the strengths of model-free advantage-weighted policy optimization and model-based reanalysis techniques.

Apart from the benchmark results, we conduct experiments under a small data setting that only has $\frac{1}{10}$th of the dataset used for benchmark experiments, and we observe that GOPlan surpasses other baselines by a significant margin. Specifically, GOPlan delivers an average improvement of 23% compared to the best-performing baseline CRL, and 38% compared to WGCSL. It is noteworthy that GOPlan alone achieves an average return over 10 on the challenging HandReach-s task, while other baselines fail on the HandReach-s task with small data budget and high-dimensional states.

Table 1: Average return with standard deviation on the offline goal-conditioned benchmark and the small dataset setting.

| Task | GOPlan | BC | GCSL | WGCSL | GEAW | AM | CRL | g-TD3-BC | g-TT | g-DT |
|---|---|---|---|---|---|---|---|---|---|---|
| PointReach | $46.09_{\pm0.1}$ | $39.36_{\pm0.4}$ | $39.27_{\pm0.4}$ | $44.40_{\pm0.1}$ | $42.95_{\pm0.1}$ | $43.56_{\pm0.6}$ | $44.53_{\pm0.2}$ | $43.69_{\pm0.2}$ | $45.39_{\pm0.5}$ | $45.85_{\pm0.5}$ |
| PointRooms | $43.57_{\pm1.8}$ | $33.17_{\pm0.5}$ | $33.05_{\pm0.5}$ | $36.15_{\pm0.8}$ | $36.02_{\pm0.5}$ | $33.45_{\pm1.9}$ | $40.12_{\pm1.0}$ | $42.32_{\pm0.8}$ | $42.46_{\pm0.8}$ | $43.20_{\pm1.2}$ |
| Reacher | $40.67_{\pm0.3}$ | $35.72_{\pm0.3}$ | $36.42_{\pm0.3}$ | $40.57_{\pm0.2}$ | $38.89_{\pm0.1}$ | $37.48_{\pm4.1}$ | $37.79_{\pm0.2}$ | $37.39_{\pm0.3}$ | $38.79_{\pm0.3}$ | $38.65_{\pm0.4}$ |
| SawyerReach | $40.43_{\pm0.3}$ | $32.91_{\pm0.3}$ | $33.65_{\pm0.3}$ | $40.12_{\pm0.2}$ | $37.42_{\pm0.3}$ | $40.91_{\pm0.2}$ | $36.73_{\pm0.3}$ | $30.96_{\pm1.7}$ | $40.25_{\pm0.3}$ | $41.10_{\pm0.3}$ |
| SawyerDoor | $44.42_{\pm0.3}$ | $35.03_{\pm0.2}$ | $35.67_{\pm0.1}$ | $42.81_{\pm0.2}$ | $40.03_{\pm0.1}$ | $42.49_{\pm0.5}$ | $38.58_{\pm0.4}$ | $35.90_{\pm0.5}$ | $44.32_{\pm0.4}$ | $44.23_{\pm0.3}$ |
| FetchReach | $47.33_{\pm0.2}$ | $42.03_{\pm0.2}$ | $41.72_{\pm0.3}$ | $46.33_{\pm0.0}$ | $45.01_{\pm0.1}$ | $46.50_{\pm0.1}$ | $46.10_{\pm0.1}$ | $45.51_{\pm0.3}$ | $47.15_{\pm0.1}$ | $47.03_{\pm0.2}$ |
| FetchPush | $39.15_{\pm0.6}$ | $31.56_{\pm0.6}$ | $28.56_{\pm0.9}$ | $39.11_{\pm0.1}$ | $37.42_{\pm0.2}$ | $30.49_{\pm2.1}$ | $36.52_{\pm0.6}$ | $30.83_{\pm0.6}$ | $38.90_{\pm0.8}$ | $38.63_{\pm0.7}$ |
| FetchPick | $37.01_{\pm1.1}$ | $31.75_{\pm1.2}$ | $25.22_{\pm0.8}$ | $34.37_{\pm0.5}$ | $34.56_{\pm0.5}$ | $34.07_{\pm0.6}$ | $35.77_{\pm0.2}$ | $36.51_{\pm0.5}$ | $36.65_{\pm0.8}$ | $35.24_{\pm1.0}$ |
| FetchSlide | $10.08_{\pm0.8}$ | $0.84_{\pm0.3}$ | $3.05_{\pm0.6}$ | $10.73_{\pm1.0}$ | $4.55_{\pm1.7}$ | $6.92_{\pm1.2}$ | $9.91_{\pm0.2}$ | $5.88_{\pm0.6}$ | $9.61_{\pm1.0}$ | $8.20_{\pm1.7}$ |
| HandReach | $28.28_{\pm5.3}$ | $0.06_{\pm0.1}$ | $0.57_{\pm0.6}$ | $26.73_{\pm1.2}$ | $0.81_{\pm1.5}$ | $0.02_{\pm0.0}$ | $6.46_{\pm2.0}$ | $5.21_{\pm1.6}$ | $24.12_{\pm1.4}$ | $22.80_{\pm1.5}$ |
| **Average** | **37.70** | 28.24 | 27.71 | 36.13 | 31.76 | 31.58 | 33.25 | 31.42 | 36.76 | 36.49 |
| FetchPush-s | $37.31_{\pm0.5}$ | $25.54_{\pm1.0}$ | $26.30_{\pm0.7}$ | $32.35_{\pm0.9}$ | $33.68_{\pm1.9}$ | $32.93_{\pm0.6}$ | $31.72_{\pm1.5}$ | $30.92_{\pm0.6}$ | $37.02_{\pm1.1}$ | $37.10_{\pm0.9}$ |
| FetchPick-s | $32.85_{\pm0.3}$ | $23.05_{\pm1.0}$ | $23.71_{\pm1.4}$ | $29.12_{\pm0.2}$ | $30.92_{\pm0.5}$ | $25.56_{\pm3.5}$ | $32.27_{\pm0.8}$ | $29.06_{\pm4.0}$ | $32.53_{\pm1.5}$ | $31.23_{\pm1.2}$ |
| FetchSlide-s | $5.04_{\pm0.4}$ | $0.31_{\pm0.1}$ | $0.98_{\pm0.4}$ | $0.22_{\pm0.1}$ | $0.30_{\pm0.1}$ | $1.97_{\pm2.7}$ | $4.74_{\pm0.6}$ | $0.16_{\pm0.2}$ | $2.39_{\pm1.1}$ | $1.60_{\pm1.4}$ |
| HandReach-s | $10.11_{\pm1.4}$ | $0.16_{\pm0.1}$ | $0.13_{\pm0.1}$ | $0.12_{\pm0.1}$ | $0.03_{\pm0.0}$ | $0.08_{\pm0.1}$ | $0.45_{\pm0.3}$ | $1.6_{\pm2.3}$ | $4.22_{\pm0.6}$ | $5.69_{\pm1.2}$ |
| FetchPush-es | $18.91_{\pm1.2}$ | $10.57_{\pm1.9}$ | $5.96_{\pm2.1}$ | $13.25_{\pm2.8}$ | $10.35_{\pm2.3}$ | $11.28_{1.6}$ | $16.13_{\pm1.8}$ | $6.27_{\pm0.9}$ | $15.76_{\pm1.2}$ | $16.56_{\pm1.4}$ |
| FetchPick-es | $14.19_{\pm0.9}$ | $4.16_{\pm2.0}$ | $1.67_{\pm0.8}$ | $6.49_{\pm1.7}$ | $3.17_{\pm0.3}$ | $4.16_{1.3}$ | $13.12_{\pm0.8}$ | $5.96_{\pm2.2}$ | $12.28_{\pm1.4}$ | $11.82_{\pm1.4}$ |
| **Average** | **19.73** | 10.63 | 9.79 | 13.59 | 13.07 | 12.66 | 16.40 | 12.33 | 17.37 | 17.33 |

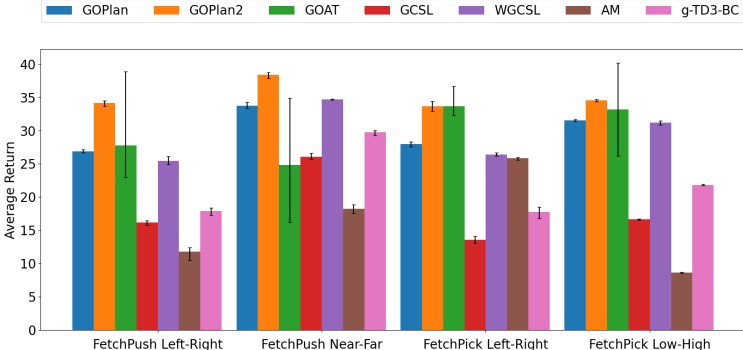

Figure 4: Average performance on OOD generalization tasks over 5 random seeds. The error bars depict the upper and lower bounds of the returns within each task group.

## 5.2 Generalization to OOD Goals

We also conducted experiments on the four OOD generalization task groups in Figure 4. The results demonstrate that GOPlan outperforms other baselines. Specifically, by planning actions for OOD goals, GOPlan2 showcases significant improved performance, outperforming WGCSL by nearly 20% on average and enjoying less variance across IID and OOD tasks than recent SOTA method GOAT (Yang et al., 2023) for OOD goal generalization. The full results of this experiment can be found in Appendix D.1. The results obtained from our experiments provide empirical evidence in support of the effectiveness of using dynamics models for enhancing generalization towards achieving OOD goals.

## 5.3 Robustness to Noise

In this experimental setup, we investigated the robustness of various algorithms in modified FetchReach environments, which are subjected to different levels of Gaussian action noise applied to the policy action outputs. The datasets for these environments were obtained from Ma et al. (2022). As shown in Figure 5, GOPlan exhibits robust performance in this setup even under a noise level of 1.5.

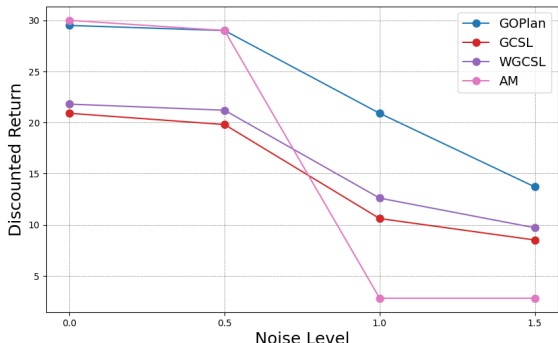

Figure 5: Stochastic environment evaluation.

## 5.4 Finetuning with Different Models

In offline GCRL, the reward function is always accessible for the agent, so we do not need to learn a reward model. However, it is feasible to learn a world model, including the transitions and the rewards. We design two variants: *GOPlan with reward models*, in which we do not use the environment-defined reward function, and instead train a reward model using the offline dataset with a binary cross entropy loss to predict the sparse reward; *GOPlan with RSSM*, in which we replace the environment-defined reward function and our learned transition models with the RSSM world model used in Dreamer (Hafner et al., 2019; 2020). Since the inference of the RSSM model consists of a stochastic latent code, we directly indicate the uncertainty by its standard deviation. In Figure 2, we evaluate the two variants with GOPlan in FetchPush, FetchPick and SawyerDoor. We found that GOPlan with reward models shows a similar performance as the vanilla GOPlan. However, GOPlan with RSSM shows slight performance decline.

| | FetchPush | FetchPick | SawyerDoor |
|---|---|---|---|
| GOPlan | $39.15_{\pm 0.6}$ | $37.01_{\pm 1.1}$ | $44.42_{\pm 0.3}$ |
| GOPlan with reward models | $38.75_{\pm 0.5}$ | $37.10_{\pm 1.0}$ | $44.32_{\pm 0.3}$ |
| GOPlan with RSSM | $38.20_{\pm 0.4}$ | $36.40_{\pm 0.8}$ | $43.20_{\pm 0.5}$ |

Table 2: Comparsions between GOPlan, GOPlan with learned reward models, and GOPlan with RSSM.

## 5.5 Ablations

This section investigates the impact of different components on the performance of GOPlan.

**Different prior policy for planning.** Initially, we compare different models as the prior policy for planning. We denote planning with model "$X$" as "$X-$plan", and advantage-weighted CGAN as "ACGAN". Based on the results in Table 3, GOPlan2 and ACGAN-plan exhibit superior performance over Gaussian and VAE models, owing to the performant OOD action avoidance capabilities, which are advantageous for long-term planning. Furthermore, policies that incorporate advantage weighting, including ACGAN, Weighted CVAE, and Weighted Gaussian, have shown superior performance compared to their unweighted counterparts. This underscores the efficacy of the advantage-weighted policy approach and its potential for application in planning scenarios.

**Ablation on the two stages of GOPlan.** Next, we compare the performance of three different variants of GOPlan to assess the significance of the pretraining and finetuning stages in GOPlan. The variants include: (1) ACGAN (i.e., GOPlan without finetuning), (2) CGAN with $w(s, a, g) = 1$ in Eq.equation 7, (3) ACGAN-plan with model-based planning with action candidates from ACGAN. Results presented in Figures 6 (a)(b) demonstrate that ACGAN consistently outperforms CGAN in terms of average return. Furthermore, ACGAN-plan is stable and efficient, but it imposes excessive computation for online interaction

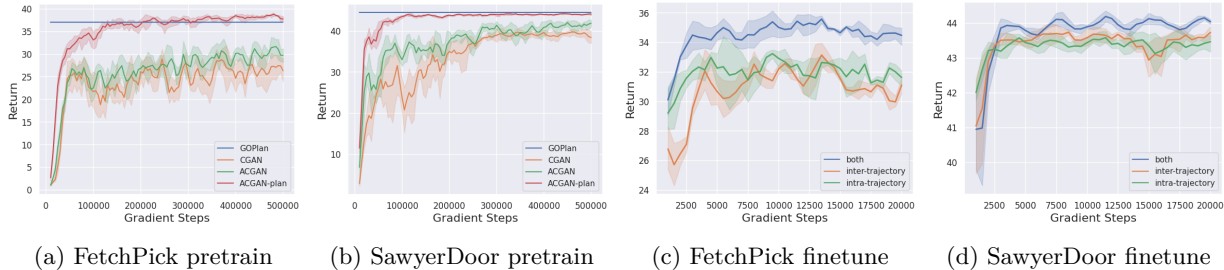

|    (a) FetchPick pretrain    |    (b) SawyerDoor pretrain    |    (c) FetchPick finetune    |    (d) SawyerDoor finetune    |

Figure 6: Ablation studies of GOPlan. The static lines in (a) and (b) are the convergent performance of GOPlan, i.e., ACGAN after finetuning.

Table 3: Comparison of different prior policies for planning.

| Algorithm | FetchPick | FetchPush | SawyerDoor |
|---|---|---|---|
| GOPlan2 | $\mathbf{38.20}_{\pm 0.8}$ | $\mathbf{39.75}_{\pm 0.8}$ | $\mathbf{45.13}_{\pm 0.3}$ |
| ACGAN-plan | $\mathbf{37.01}_{\pm 1.1}$ | $38.42_{\pm 0.9}$ | $\mathbf{44.42}_{\pm 0.3}$ |
| CGAN-plan | $33.95_{\pm 2.3}$ | $36.51_{\pm 2.6}$ | $42.04_{\pm 1.0}$ |
| weighted CVAE-plan | $35.02_{\pm 0.9}$ | $\mathbf{39.11}_{\pm 0.9}$ | $42.88_{\pm 0.7}$ |
| CVAE-plan | $30.32_{\pm 1.2}$ | $34.36_{\pm 1.1}$ | $39.73_{\pm 0.7}$ |
| weighted Gaussian-plan | $36.41_{\pm 0.6}$ | $38.62_{\pm 0.9}$ | $43.71_{\pm 0.8}$ |
| Gaussian-plan | $29.95_{\pm 1.0}$ | $30.97_{\pm 1.5}$ | $36.61_{\pm 0.5}$ |

(30 Hz on GTX 3090). After finetuning ACGAN with our reanalysis methods, GOPlan achieves comparable performance as ACGAN-plan, and offers faster interaction (500 Hz).

**Inter-trajectory and intra-trajectory reanalysis.** During finetuning, both inter-trajectory and intra-trajectory reanalysis techniques are used in GOPlan. To assess the effectiveness of each approach, we compare the individual impacts and examine their joint influence on performance, as illustrated in Figures 6 (c)(d). Specifically, GOPlan generates 50% trajectories for each reanalysis, while "inter-trajectory"(or "intra-trajectory") performs 100% inter-trajectory (or intra-trajectory) reanalysis. Our findings are threefold: (1) inter-trajectory reanalysis can introduce a distribution shift that initially decreases performance, while intra-trajectory reanalysis has no such effect. (2) Both intra and inter-trajectory reanalysis lead to considerable improvement over the pretrained policy after sufficient finetuning steps. (3) The two reanalysis methods can work synergistically to yield the best performance.

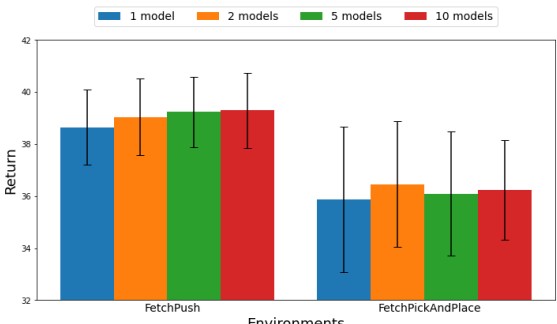

Figure 7: Ablation studies of GOPlan on different ensemble sizes.

**Ensemble sizes and uncertainty truncation.** We investigates the influence of different ensemble sizes of the dynamics models on the performance of GOPlan. We use $1, 2, 5, 10$ dynamics models in the finetuning

stage to generate data for reanalysis. The results, presented in Figure 7, demonstrate the robustness of GOPlan to varying ensemble sizes, with all ensemble sizes exceeding the performance of a single dynamics model. This observation underscores the significance of uncertainty qualification through ensembles for GOPlan's performance. Notably, when only one model is employed, the inability to estimate uncertainty for truncating trajectories with high uncertainty leads to a degradation in performance. Furthermore, in the FetchPush task, where the variance in performance is relatively small, the average performance exhibits a stable improvement as the number of models increases.

## 6 Conclusions and Discussions

This paper proposes GOPlan, a novel model-based offline GCRL algorithm designed to effectively learn general goal-reaching policies from diverse and multi-goal offline datasets. GOPlan comprises two components: pretraining a prior policy using an advantage-weighted CGAN and finetuning the policy with reanalysis. The advantage-weighted CGAN exhibits distinct mode separation and enhances the action distribution based on advantage values, thereby mitigating the OOD action issue and promoting long-term planning for offline GCRL. In addition, the reanalysis method generates high-performing and low-uncertainty imaginary samples by planning with learned models towards both intra-trajectory and inter-trajectory goals. Experimental results demonstrate that GOPlan achieves SOTA performance on offline GCRL benchmark tasks. Importantly, GOPlan yields greater improvements for challenging settings with limited data and OOD goal generalization, highlighting its potential advantages for practical scenarios.

Despite its promising features, GOPlan could be enhanced to efficiently learn from human demonstrations, which exhibit more diverse patterns than our testing benchmarks (Shafiullah et al., 2022; Lynch et al., 2019). Future research directions also include extending GOPlan to high-dimensional setting, which may require planning with advanced world models as proposed in (Janner et al., 2021; Wang et al., 2023; Janner et al., 2022), as well as planning in latent spaces as studied in (Hafner et al., 2019; Nguyen et al., 2021). In addition, researchers should consider its potential ethical issues when applying the technique to real-world applications, such as the misuse of technology and the safety in scenarios where mistakes could be hazardous.

**Acknowledgements**

We would like to thank Yue Jin, the editors and the reviewers for their comments to improve the paper. We also express our gratitude to Tristan Tomilin for his contribution through his presentation on this paper at the Goal-conditioned RL workshop at NeurIPS 2023. GM acknowledges support from UKRI Turing AI Acceleration Fellowship (EPSRC EP/V024868/1).

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

## A    Model-based Planning

In GOPlan's intra-trajectory and inter-trajectory reanalysis, GOPlan invokes a function named `Plan`, which is a model-based planning method, that employs a model to simulate multiple imaginary trajectories, assigns scores to the initial actions, and generates the final action based on these scores. A variety of the model-based planning methods could be used here, such as model-predictive control (MPC) (Nagabandi et al., 2020) and Monte-Carlo tree search (MCTS) (Schrittwieser et al., 2021). In our implementation, we use a similar planning algorithm as PlanGAN (Charlesworth & Montana, 2020), as demonstrated in Algorithm 2. Our planning method can be refined by including a value function that estimates the return at horizon $T$ (Argenson & Dulac-Arnold, 2021), which we leave for future work.

---

**Algorithm 2** Model-based Planning.

---

**Initialise:** $N$ dynamics models $\{M_i\}_{i=1}^N$, policy $\pi$; current state $s_0$, goal $g$, reward function $r$.

1: **for** $c = 1, ..., C$ **do**
2:     $z \sim \mathcal{N}(0, 1)$
3:     $a_0^c = \pi(s_0, g, z)$                                      ▷ Sample $C$ initial actions $\{a_0^c\}_{c=1}^C$.
4:     $i \sim \text{Uniform}(1, \ldots, N)$
5:     $\hat{s}_1^c = M_i(s_0, a_0^c)$                                  ▷ Predict $C$ next states $\{\hat{s}_1^c\}_{c=1}^C$.
6:     **for** $h = 1, ..., H$ **do**
7:         $\hat{s}_{h,1}^c = \hat{s}_1^c$                               ▷ Duplicate every next state $H$ times.
8:         **for** $k = 1, ..., K$ **do**
9:             $z \sim \mathcal{N}(0, 1)$
10:             $a_{h,k}^c = \pi(\hat{s}_{h,k}^c, g, z)$
11:             $i \sim \text{Uniform}(1, \ldots, M)$
12:             $\hat{s}_{h,k+1}^c = M_i(\hat{s}_{h,k}^c, a_{h,k}^c)$            ▷ Generate $H$ trajectories of $K$ steps.
13:         **end for**
14:         $R_{c,h} = \sum_{k=0}^K r(\hat{s}_{h,k}^c, a_{h,k}^c, g)$
15:     **end for**
16:     $R_c = \frac{1}{H} \sum_{h=1}^H R_{c,h}$                          ▷ Average all cumulative returns.
17:     $R_c = \frac{R_c}{\sum_{c=1}^C R_c}$                         ▷ Normalize all cumulative returns.
18: **end for**
19: $a^* = \frac{\sum_{c=1}^C e^{\kappa R_c} \cdot a_0^c}{\sum_{c=1}^C e^{\kappa}}$                      ▷ Exponentially weight the actions.
20: **return** $a^*$

---

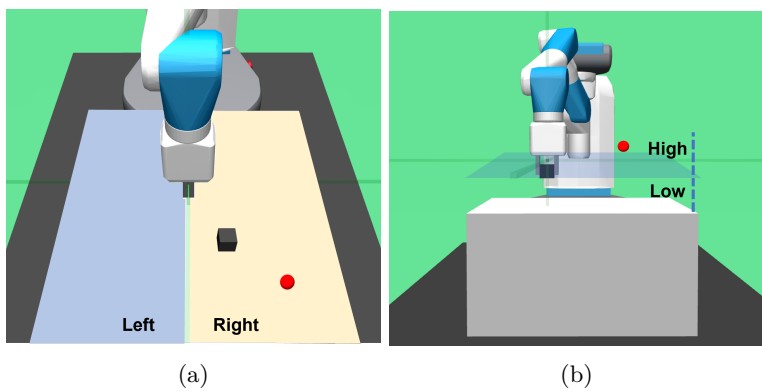

(a)                                 (b)

Figure 8: Goal-conditioned tasks for OOD generalization. (a) Push Left-Right, (b) Pick Low-High.

## B   Environments and Datasets

We use a set of manipulation and navigation environments to evaluate the algorithm, including PointReach, PointRooms, Reacher, SawyerReach, SawyerDoor, FetchReach, FetchPush, FetchPick, FetchSlide and HandReach. The environments are illustrated in Figure 9. All of the environments have continuous state space, action space and goal space. The details of the environments can be found in Appendix F in (Yang et al., 2022b). The offline datasets are collected by a pre-trained policy using DDPG and hindsight relabelling (Andrychowicz et al., 2017), where the actions from the policy are perturbed by adding Gaussian noise with zero mean and 0.2 standard deviation to increase the diversity and multi-modality of the dataset (Yang et al., 2022b).

We also introduce four groups of tasks for evaluating OOD generalization ability (Yang et al., 2023). There are four task groups: FetchPush Left-Right, FetchPush Near-Far, FetchPick Left-Right, FetchPick Low-High. We illustrate FetchPush Left-Right and FetchPick Low-High in Figure 8. In the data collection, we only

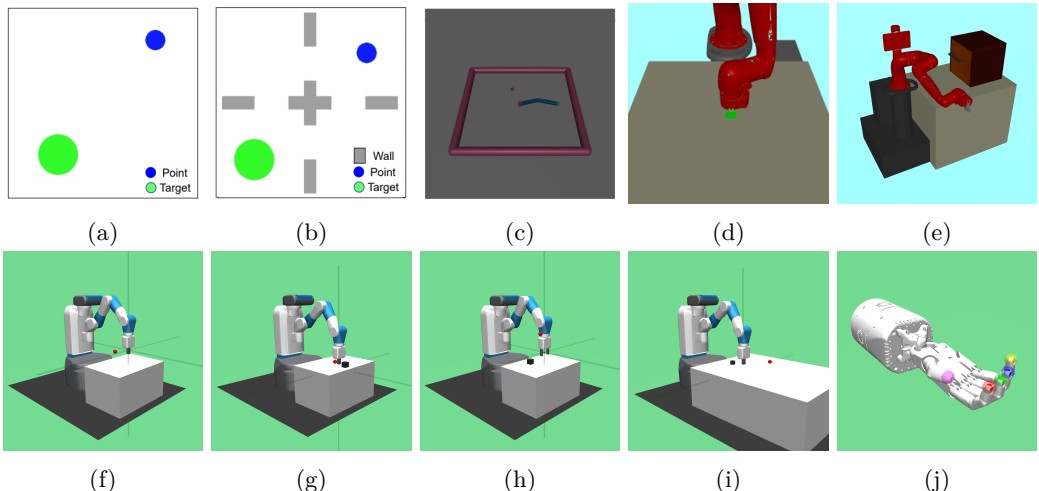

Figure 9: Goal-conditioned tasks. (a) PointReach, (b) PointRooms, (c) Reacher, (d) SawyerReach, (e) SawyerDoor, (f) FetchReach, (g) FetchPush, (h) FetchPick, (i) FetchSlide, and (j) HandReach.

store the trajectories whose achieved goals are all in the IID (independent identically distributed) region. The IID region is defined by each task group, shown in Table 5. The dataset division standard refers to the location requirements of initial states and desired goals for IID tasks (e.g., Right2Right). For OOD tasks, at least one of the initial state or the desired goal does not satisfy the IID requirement (e.g., Right2Left, Left2Right, Left2Left). We collect relatively smaller datasets of 5000 trajectories. Once we train a policy on the offline dataset, we evaluate the policy both on the IID tasks and the OOD tasks.

## C   Implementation Details

To implement GOPlan, we train a fully implicit policy, denoted as $\pi(a_t \mid s_t, g, z)$, with $z$ representing a 64-dimensional diagonal Gaussian noise. All associated models, including the value function and dynamics models, utilize 3-layer multi-layer perceptrons with 512 units in each layer. Furthermore, the policy network incorporates batch normalization, while the discriminator network employs leaky ReLU activation function. We use Adam optimizer with a learning rate of $1 \times 10^{-4}$. During the fine-tuning phase, we gather $J_{intra} = 2000$ and $J_{inter} = 2000$ trajectories for intra-trajectory reanalysis and inter-trajectory reanalysis, respectively. After each collection, the prior policy is finetuned with 500 gradient steps. This process is performed a total of $I = 10$ times. In order to show the difference in the ablation study, we use $J_{intra} = 200$ and $J_{inter} = 200$ there. Table 4 gives a list and description of them, as well as their default values.

## D   Additional Experiments

In this section, we provide more experiments to study our proposed algorithm.

### D.1   OOD Generalization Tasks

In the main paper, we present the statistical results of experiments conducted on four OOD generalization task groups (see Figure 4): FetchPush Left-Right, FetchPush Near-Far, FetchPick Left-Right, and FetchPick Low-High. The complete set of results for this experiment is shown in Table 6. Among these tasks, GOPlan with online planning (GOPlan2) demonstrates the highest performance in both IID and OOD tasks. In most of the task groups, GOPlan is comparable with the recent SOTA algorithm GOAT (Yang et al., 2023), while it outperforms previous approaches. These results validate the effectiveness of incorporating dynamics models to enhance generalization for OOD goals.

Table 4: Hyper-parameters.

| Symbol | Description | Default Value |
|---|---|---|
| $\gamma$ | discount factor | 0.98 |
| $\beta$ | coefficient in the exponential advantage weight | 60 |
| $N$ | Number of dynamics models | 3 |
| | ACGAN noise dimension | 64 |
| | Range of the exponential weight | $[0, 10]$ |
| | Batch size | 512 |
| $I$ | Finetuning episodes | 10 |
| $I_{intra}$ | # collected Intra-trajectories each episode | 1000 |
| $I_{inter}$ | # collected Inter-trajectories each episode | 1000 |
| $u$ | Uncertainty threshold | 0.1 |
| | Finetuning updates every episode | 1000 |
| | Pretraining updates | $5 \times 10^5$ |
| | Size of reanalysis buffer | $2 \times 10^6$ |
| $\kappa$ | Exponential weight | 2 |
| $K$ | Planning horizon | 20 |
| $C$ | Number of initial actions | 20 |
| $H$ | Copies of initial actions | 10 |

Table 5: Information about 4 Task Groups and Datasets.

| Datasets (Task Group) | IID region | IID task | OOD task | Dataset Division Standard |
|---|---|---|---|---|
| Push Left-Right | Right | Right2Right | Right2Left, Left2Right, Left2Left. | the object's $y$ coordinate value > the initial position |
| Push Near-Far | Near | Near2Near | Near2Far, Far2Near, Far2Far. | the $L_2$ distance between the object and the initial position $\leq 0.15$ |
| Pick Left-Right | Right | Right2Right | Right2Left, Left2Right, Left2Left. | the object's $y$ coordinate value > the initial position |
| Pick Low-High | Low | Low2Low | Low2High. | the object's $z$ coordinate value < 0.6 |

Table 6: Average returns with standard deviations on the OOD benchmark. The results of GOAT are taken from the original paper (Yang et al., 2023).

| Task Group | Task | GOPlan | GOPlan2 | GOAT | GCSL | WGCSL | AM | g-TD3-BC |
|---|---|---|---|---|---|---|---|---|
| FetchPush Left-Right | Right2Left | $27.17_{\pm 2.5}$ | $33.65_{\pm 1.5}$ | $24.9_{\pm 2.4}$ | $15.78_{\pm 1.5}$ | $25.44_{\pm 1.2}$ | $11.93_{\pm 4.1}$ | $17.73_{\pm 2.1}$ |
| | Left2Right | $26.64_{\pm 2.5}$ | $34.26_{\pm 1.3}$ | $24.3_{\pm 2.7}$ | $15.98_{\pm 1.4}$ | $25.13_{\pm 1.5}$ | $10.48_{\pm 4.8}$ | $18.35_{\pm 1.2}$ |
| | Left2Left | $26.84_{\pm 3.2}$ | $34.51_{\pm 1.7}$ | $23.0_{\pm 2.8}$ | $16.48_{\pm 0.8}$ | $24.91_{\pm 1.8}$ | $12.44_{\pm 4.7}$ | $17.31_{\pm 0.9}$ |
| | Right2Right | $26.87_{\pm 1.9}$ | $34.21_{\pm 2.1}$ | $38.9_{\pm 1.0}$ | $16.42_{\pm 1.6}$ | $26.14_{\pm 2.7}$ | $12.35_{\pm 4.0}$ | $18.12_{\pm 0.8}$ |
| | Average | 26.88 | **34.16** | **27.78** | 16.16 | 25.41 | 11.80 | 17.88 |
| FetchPush Near-Far | Far2Far | $33.32_{\pm 1.8}$ | $38.47_{\pm 0.7}$ | $16.2_{\pm 0.6}$ | $25.72_{\pm 1.3}$ | $34.72_{\pm 0.8}$ | $18.38_{\pm 4.7}$ | $29.30_{\pm 0.9}$ |
| | Near2Near | $33.59_{\pm 1.3}$ | $38.74_{\pm 1.0}$ | $34.9_{\pm 1.3}$ | $25.96_{\pm 1.1}$ | $34.73_{\pm 0.9}$ | $18.86_{\pm 3.8}$ | $29.93_{\pm 1.8}$ |
| | Near2Far | $34.26_{\pm 1.4}$ | $38.53_{\pm 0.7}$ | $25.3_{\pm 2.1}$ | $26.58_{\pm 0.9}$ | $34.69_{\pm 1.0}$ | $17.54_{\pm 4.2}$ | $29.86_{\pm 1.4}$ |
| | Far2Near | $33.79_{\pm 2.2}$ | $37.82_{\pm 0.9}$ | $23.0_{\pm 1.4}$ | $26.01_{\pm 0.5}$ | $34.65_{\pm 1.3}$ | $18.13_{\pm 4.7}$ | $30.06_{\pm 1.0}$ |
| | Average | 33.74 | **38.39** | 24.85 | 26.07 | **34.70** | 18.23 | 29.79 |
| FetchPick Left-Right | Left2Right | $27.99_{\pm 1.7}$ | $32.92_{\pm 2.1}$ | $33.4_{\pm 0.6}$ | $13.03_{\pm 2.3}$ | $26.69_{\pm 0.6}$ | $25.92_{\pm 1.7}$ | $16.78_{\pm 1.4}$ |
| | Left2Left | $28.29_{\pm 0.8}$ | $33.74_{\pm 1.6}$ | $32.3_{\pm 1.5}$ | $13.72_{\pm 1.5}$ | $26.26_{\pm 0.8}$ | $25.89_{\pm 1.8}$ | $18.33_{\pm 1.9}$ |
| | Right2Left | $27.59_{\pm 1.0}$ | $33.72_{\pm 1.5}$ | $32.3_{\pm 0.7}$ | $13.55_{\pm 1.6}$ | $26.55_{\pm 1.4}$ | $25.57_{\pm 1.4}$ | $18.54_{\pm 2.2}$ |
| | Right2Right | $28.07_{\pm 1.1}$ | $34.39_{\pm 0.8}$ | $36.7_{\pm 0.6}$ | $14.13_{\pm 2.1}$ | $26.17_{\pm 0.5}$ | $25.97_{\pm 1.1}$ | $17.53_{\pm 2.3}$ |
| | Average | 27.99 | **33.69** | **33.68** | 13.61 | 26.42 | 25.84 | 17.79 |
| FetchPick Low-High | Low2Low | $31.43_{\pm 1.5}$ | $34.37_{\pm 0.8}$ | $40.2_{\pm 0.2}$ | $16.55_{\pm 1.3}$ | $30.84_{\pm 0.8}$ | $8.60_{\pm 3.5}$ | $21.96_{\pm 2.4}$ |
| | Low2High | $31.66_{\pm 1.8}$ | $34.72_{\pm 1.1}$ | $26.2_{\pm 2.4}$ | $16.75_{\pm 0.8}$ | $31.50_{\pm 0.8}$ | $8.70_{\pm 2.7}$ | $21.73_{\pm 2.1}$ |
| | Average | 31.55 | **34.54** | **33.20** | 16.65 | 31.17 | 8.65 | 21.84 |

