# OpenReview forum: "GOPlan: Goal-conditioned Offline Reinforcement Learning by Planning with Learned Models"
_TMLR — Accepted by TMLR_

### Review · Reviewer_UJw9 · 2024-02-11

**Summary Of Contributions:**

This paper proposes GOPlan, a novel model-based offline goal-based RL method that is capable of capturing the multi-modal action distribution and dealing with out-of-distribution actions. The method has two main components:

1) Learning prior policy with an advantage-weighted CGAN and an ensemble of dynamics models that predict next state from offline data;

2) Finetune the policy with imaginary rollouts in the model. The rollouts are conducted by planning with the learned prior policy, where GOPlan first samples actions and generates many next steps, then takes the next step as weighted average of the next steps (weights are the future return over many imaginary rollouts). Specially, the imaginary rollouts have two sources: intra-trajectory or inter-trajectory, where the former tries to reach from one state in an existing trajectory in the dataset to another, and the latter tries to reach from one state in an existing trajectory to another state in another existing trajectory. The generated trajectory could be accepted, accepted with goal relabeled (for intra-trajectory), or discarded.

The method is tested on various testbeds which proves its superiority to existing offline goal-based RL methods.

**Audience:**

Yes

**Broader Impact Concerns:**

This paper does not include a broader impact statement, and since the testbed of the paper is all on simulation tasks, I do not feel there are any major broader impact concerns. In spite of this, it would be great if the author could include a broader impact statement in the revised version (see requested changes).

**Claims And Evidence:**

Yes

**Requested Changes:**

**Below is the change that is critical to securing my recommendation for acceptance:**

More carefully discuss in related work and preferably compare with world model methods such as Dreamer in the experiments.

**Below are the changes that strengthen the work:**

- Include decision diffuser [1] (and possibly decision transformer [2]) as baseline. Decision diffuser is a newer solution superior to trajectory transformer, is generative-model-based, and can work with goal-based RL, which corresponds to “satisfying constraints” in the decision diffuser paper.
- Discuss the possibility of using other generative models, such as normalizing flow and diffusion models as the agent to address the multimodal issue;

- Check the math symbols appear in the paper and make sure they are properly introduced (specifically, see weakness point 1);

- Include a discussion about the limitation of the proposed method, and preferably also an impact statement. For example, while this work mostly deals with simulated environments and thus does not have much direct impact on real-world applications, it is generally an automation of decision-making that could lead to loss of job, abused use of technology and human deskilling. Besides, hallucination of the generated trajectory caused by inaccurate modeling could also be harmful in some scenarios where making mistakes is dangerous.

**References:**

[1] Ajay, Anurag, et al. "Is conditional generative modeling all you need for decision-making?." In ICLR, 2023.

[2] Chen, Lili, et al. "Decision transformer: Reinforcement learning via sequence modeling." In NeurIPS, 2021.

**Strengths And Weaknesses:**

**Strengths:**

- Generally, the paper is well-written with its ideas clearly conveyed. The illustrations well convey the high-level framework of GOPlan (Fig. 1) and possible outcomes of reanalysis (Fig. 3).  The pseudocode is also quite helpful, making the procedure easy to follow. The hyperparameters and experimental settings are clearly listed in the supplementary materials. Some details, though, could be improved; see “weakness” section, point 1 for advice.

- The proposed idea, specifically, the reanalysis part, is quite interesting. The inter- and intra-trajectory selection of rollout effectively and actively builds up a topological graph between states that appear in the dataset, which is an elegant design.

- The results are convincing with extensive ablation study; the ablations in Sec. 5.2 to 5.4 clearly show the robustness of the proposed method against noise, out-of-distribution goals, and small dataset. The effectiveness of the dynamic model ensemble is also examined.

**Weaknesses:**

- Some details of the presentation of this paper could be improved. For example, the terms $l^p, l^v, l^r$ in Eq. 2 are not properly introduced.

- There is neither limitation nor broader impact statement included in the paper.

- Though this paper briefly mentions world models in the beginning, it does not discuss in detail the feasibility of actually learning a world model instead of only dynamics model (that is, learning everything including transitions and reward), and neither cite or compare with representative world model papers such as Dreamerv3 [1].

- (This is not a limitation; it is more of a question:) Why is case (a-2) of the intra-trajectory reanalysis leading to discarding the new trajectory? Is it possible to include the trajectory but with a relabeled goal as what the method does in case (b-2)?

**References:**

[1] Hafner, D., Pasukonis, J., Ba, J., & Lillicrap, T. Mastering diverse domains through world models. arXiv, 2023.

---

> ### Author Response · Authors · 2024-04-03
>
> Thank you sincerely for the effort and time you spent reviewing our manuscript. We have taken care to address each point you raised, making revisions highlighted in blue within the revised paper. We believe these adjustments will effectively alleviate your concerns.
>
> **Presentation (W1 & C3)**
> In continuous control settings, the loss functions $l^p(\cdot, \cdot)$, $l^v(\cdot, \cdot)$, $l^r(\cdot, \cdot)$ are mean square error (MSE) between the inputs. We have corrected this in the revised version.
>
> **Limitations and Impacts (W2 & C4)**
> Thanks for reminding us to discuss the limitations and the broader impact. We have explained them in the last section of the revised paper.
>
> **World Models (W3)**
> In offline GCRL, the reward function is always accessible for the agent, so we do not need to learn a reward model. However, it is feasible to learn a world model, including the transitions and rewards. To solve the concerns, we design 2 variants:
>
> 1. **GOPlan with reward models**, in which we do not use the environment-defined reward function, and instead train a reward model using the offline dataset with a binary cross entropy loss to predict the sparse reward;
> 2. **GOPlan with RSSM**, in which we replace the environment-defined reward function and our learned transition models with the RSSM world model used in Dreamer (Hafner et al., 2019 and 2020). Since the inference of the RSSM model consists of a stochastic latent code, we directly indicate the uncertainty by its standard deviation.
>
> We evaluate these two variants with GOPlan in FetchPush, FetchPick, and SawyerDoor. We found that GOPlan with reward models shows a similar performance as the GOPlan with environment-defined reward function. However, GOPlan with RSSM shows slight performance decline. This decline might be attributed to the design of RSSM, which is tailored for processing high-dimensional image inputs, while our experiments primarily utilize low-dimensional state inputs. Due to this mismatch, the high capabilities of RSSM do not confer a benefit in our specific tasks, leading to inferior performance compared to a simpler dynamics model.
>
> |                                  | FetchPush          | FetchPick          | SawyerDoor         |
> |----------------------------------|--------------------|--------------------|--------------------|
> | GOPlan                           | 39.15±0.6          | 37.01±1.1          | 44.42±0.3          |
> | GOPlan with reward models        | 38.75±0.5          | 37.10±1.0          | 44.32±0.3          |
> | GOPlan with RSSM                 | 38.20±0.4          | 36.40±0.8          | 43.20±0.5          |
>
> *Table: Comparisons between GOPlan, GOPlan with learned reward models, and GOPlan with RSSM.*
>
> **Intra-reanalysis (W4)**
> This is a very good question. We choose not to include the imagined trajectory due to two reasons:
>
> - The majority of the imagined trajectories in (a-2) can reach the goal (96% in FetchPush, 95% in FetchPick, and 97% in SawyerDoor), since the prior policy can learn to reach the goal with successful reference trajectories in the dataset.
> - The quality of the imagined trajectory cannot be guaranteed, as the imagined trajectory is generated towards another goal rather than the relabelled goal; the trajectory could be sub-optimal towards the relabeled goal.
>
> In order to solve the concern, we made experiments in FetchPush, FetchPick, and SawyerDoor with/without the relabelled trajectories. There is no evidence to show it helps fine-tune the prior policy.
>
> |                                  | FetchPush          | FetchPick          | SawyerDoor         |
> |----------------------------------|--------------------|--------------------|--------------------|
> | With relabeled trajectories      | 39.24±0.5          | 36.45±1.7          | 43.92±0.3          |
> | Without relabeled trajectories   | 39.15±0.6          | 37.01±1.1          | 44.42±0.3          |
>
> *Table: The results of two variant methods to solve scenario (a-2) in intra-reanalysis.*
>
> **Baselines (C1)**
> Thanks for your suggestions. We have added the results from the decision transformer as a baseline in the revised paper.
>
> **Generative Models (C2)**
> In theory, both normalizing flow and diffusion models can tackle the issue of multi-modality by progressively transitioning from an initial distribution to a complex target distribution, a process described by Wang et al. (2023) and Akimov et al. (2023). However, this transformation is both time-consuming and computationally demanding, rendering their inference processes inefficient. For real-world applications, there is a preference for policies that offer rapid responses, thus GAN, whose frequency can achieve 500Hz, is a more ideal solution.

---

> ### Author Response · Authors · 2024-04-03
> **Official Comment by Authors (cont'd)**
>
> **References**
>
> - Akimov, D., Kurenkov, V., Nikulin, A., Tarasov, D., and Kolesnikov, S. (2022). Let offline RL flow: Training conservative agents in the latent space of normalizing flows. arXiv preprint arXiv:2211.11096.
> - Hafner, D., Lillicrap, T., Ba, J., and Norouzi, M. (2020). Dream to control: Learning behaviors by latent imagination. In *International Conference on Learning Representations*.
> - Hafner, D., Lillicrap, T., Fischer, I., Villegas, R., Ha, D., Lee, H., and Davidson, J. (2019). Learning latent dynamics for planning from pixels. In *International Conference on Machine Learning*.
> - Wang, Z., Hunt, J. J., and Zhou, M. (2023). Diffusion policies as an expressive policy class for offline reinforcement learning. In *International Conference on Learning Representations*.

---

### Review · Reviewer_bUdD · 2024-03-07

**Summary Of Contributions:**

This paper proposes GOPlan, a new model-based method for goal-conditioned offline RL setting. This method has two training stages, and use a number of different techniques to reach the final strong performance, including planning, training with imgined trajectories, using ensemble-based uncertainty estimation to avoid OOD data. The proposed method achieves a new SOTA performance on a number of benchmark tasks and settings.

**Audience:**

Yes

**Claims And Evidence:**

Yes

**Requested Changes:**

- The anonymous code link in appendix is expired, can you fix it?
- Can you provide a discussion on computation overhead comparing the proposed method and other baselines?

**Strengths And Weaknesses:**

Strengths:
- overall the paper is written clearly and presented well.
- extensive experiment results and ablations are provided to support the empirical claims.
- combining a number of technical components and design for the goal-conditioned setting can be novel
- results look strong


Weaknesses:
- some of the components of the proposed method are from previous works and not new
- the proposed method consists of a number of components and can be more complex to use and can have a larger computation overhead compared to alternatives
- would be good to see a discussion on how much additional computation cost does the proposed method bring in. Is it significantly slower than some of the baselines you compared to?


Some questions:
- Figure 2 is interesting, however, without quantification of the results, it is a little unclear to me how much advantage do we get from using CGAN compared to CVAE, and also do we still have such a behavior in the high-dimension state-action space setting? And when you compare them do these models have the same capacity, is the comparison fair?
- Maybe I missed sth, I am a little not convinced how important is the multi-modal action distribution problem in goal-conditioned RL, can you remind me what is the main evidence this is critical? If I have a simpler network but just higher capacity will it achieve the same results?

---

> ### Author Response · Authors · 2024-04-03
>
> We are grateful for your time in evaluating our manuscript. We understand that the main concerns are focused on six key aspects. Each of the aspects raised has been individually addressed, and the manuscript has been accordingly updated in blue to reflect these changes. We hope that our responses will effectively resolve your concerns.
>
> **Multi-Modality in GCRL (Q2)** In offline GCRL, there are generally many trajectories from a state to a goal, which can present multiple counteracting action labels and even impede learning (Yang et al, 2022a; Lynch 2020). Previous studies have found that failure to solve the multi-modality problem can reduce the performance (Chen et al., 2022; Cai et al., 2022). For instance, a deterministic policy or a Gaussian policy tend to output the mean of the action distribution, leading to the mismatch between the dataset and true state-action visitation of the learned policy. The ability to learn a multi-modal action distribution is not related to the capacity of the network.
>
> **Novelty (W1)** Our first contribution is the advantage weighted CGAN (ACGAN). We propose the ACGAN to learn the highly-rewarded parts of the behaviour policy's distribution (as shown in Figure 2). Although conditioned GAN has been introduced in offline RL, they are usually used as a regularizer to keep the learned policy close to the behaviour policy without considering the reward (Chen et al., 2024; Yang et al., 2022b). The second contribution is the multi-goal reanalysis. We designed six trajectory generation methods for the scenarios that may happen in GCRL. They increase the diversity of the dataset and provide high-quality data for finetuning the policy. Previous reanalysis method (Schrittwieser et al., 2021) does not consider these requirements in GCRL.
>
> **CGAN and CVAE (Q1)** Although both CGAN and CVAE are able to learn a multi-modal action distribution, the following evidences show that CGAN outperforms CVAE:
> - In Figure 2, compared to CGAN, CVAE are prone to generate data points between the modes, which could result in large extrapolation error in offline RL.
> - In Table 2, we execute model-based planning with CGAN (CGAN-plan) and CVAE (CVAE-plan) as prior policies. The evaluation results show that CGAN-plan is 11.97%, 6.26%, 5.81% better than CVAE-plan in the FetchPick, FetchPush, and SawyerDoor environments, respectively. The results demonstrate that CGAN with model-based planning could generate better actions for reanalysis.
>
> HandReach has a high-dimension state-action space, whose state space has 62 dimensions and action space has 19 dimensions. Table 1 shows that GOPlan performs significantly better than the others in HandReach. We are interested in testing GOPlan on other high-dimensional settings and even visual-input tasks. However, due to insufficient benchmarks for now, we would like to include GOPlan's results as baselines in future works.
>
> **Comparison Fairness (Q1)** We tried our best to make fair comparisons. We keep the same capacity for all models: the generative policy in CGAN is a 3-layer neural network, each of which has 512 neurons, followed by a BatchNorm layer and a ReLU activation function, whereas the CVAE has the same architectures but without the BatchNorm layer, since we found CVAE without the layer performs better. More details can be found in the anonymous code repository.
>
> **Computation (W2, W3 & C2)** The computation overhead of training GOPlan consists of two parts: learning priors and finetuning the policy. Learning priors include training the generative policy, the discriminator, the value function, and the ensemble of dynamics, which require around 60 minutes on GTX 3090 to finish this stage (500,000 updates and 3 dynamics in the ensemble). The finetuning stage requires around 10 minutes to generate 1000 intra-trajectories and around 5 minutes for 1000 inter-trajectories, thus total 150 minutes for finetuning the policy. Although GOPlan spends more computation than WGCSL (60 min) and TD3-BC (60 min), it spends less than Trajectory Transformer (520 min with official implementation) and achieves a higher score.
>
> **Expired Repo (C1)** Thanks for pointing this out. We have uploaded the anonymous code again. Please check the new anonymous repository [https://anonymous.4open.science/r/GOPlan-0531](https://anonymous.4open.science/r/GOPlan-0531).

---

> ### Author Response · Authors · 2024-04-03
> **Official Comment by Authors (cont'd)**
>
> **References**
>
> Cai, Y., Zhang, C., Zhao, L., Shen, W., Zhang, X., Song, L., Bian, J., Qin, T., and Liu, T. (2022). TD3 with Reverse KL Regularizer for Offline Reinforcement Learning from Mixed Datasets. arXiv preprint arXiv:2212.02125.
>
> Chen, J., Ganguly, B., Xu, Y., Mei, Y., Lan, T., and Aggarwal, V. (2024). Deep Generative Models for Offline Policy Learning: Tutorial, Survey, and Perspectives on Future Directions. arXiv preprint arXiv:2402.13777.
>
> Chen, X., Ghadirzadeh, A., Yu, T., Wang, J., Gao, A. Y., Li, W., Bin, L., Finn, C., and Zhang, C. (2022). LAPO: Latent-Variable Advantage-Weighted Policy Optimization for Offline Reinforcement Learning. In *Advances in Neural Information Processing Systems*.
>
> Lynch, C., Khansari, M., Xiao, T., Kumar, V., Tompson, J., Levine, S., and Sermanet, P. (2020). Learning latent plans from play. In *Conference on Robot Learning*.
>
> Schrittwieser, J., Hubert, T. K., Mandhane, A., Barekatain, M., Antonoglou, I., and Silver, D. (2021). Online and Offline Reinforcement Learning by Planning with a Learned Model. In *Advances in Neural Information Processing Systems*.
>
> Yang, R., Lu, Y., Li, W., Sun, H., Fang, M., Du, Y., Li, X., Han, L., and Zhang, C. (2022a). Rethinking Goal-Conditioned Supervised Learning and Its Connection to Offline RL. In *International Conference on Learning Representations*.
>
> Yang, S., Wang, Z., Zheng, H., Feng, Y., and Zhou, M. (2022b). A Behavior Regularized Implicit Policy for Offline Reinforcement Learning. arXiv preprint arXiv:2202.09673.

---

### Review · Reviewer_5fHZ · 2024-03-15

**Summary Of Contributions:**

This paper presents a new method for goal-conditioned offline reinforcement learning that combines a multi-modal policy distribution with an MCTS-based approach for fine-tuning. Experimental results measure the approach's performance on a variety of continuous control environments and illustrate that the approach is competitive with the next-best approach on every task. Ablations measure the relative performance of a variety of the approach's components (most of the main components) and confirm the efficacy of each of the main components investigated.

**Audience:**

Yes

**Broader Impact Concerns:**

I have no broader impact concerns.

**Claims And Evidence:**

Yes

**Requested Changes:**

The first two weaknesses are my main concerns. While I would recommend acceptance in the current state, they would significantly strengthen the work in my view.

**Strengths And Weaknesses:**

# Strengths
- The main claims are supported by fairly comprehensive experimental results
- The method's motivation is straightforward.
- Fig. 2 is helpful in understanding the nature of the policy distributions.
- Fig. 3 is, overall, very helpful to help the reader understand most of the intuition behind the reanalysis method.
- The technical description is overall, fairly clear. The algorithmic pseudocode blocks are helpful.

# Weaknesses
- It appears that in 13/14 tasks, the difference between the method's average performance and the next-best method's average performance is smaller than the sum of their performance standard deviations, i.e., there is no environment in which the method's performance improvement over the next-best method is inarguably substantial. One explanation could be that, in 13/14 tasks, there isn't substantial room for improvement over the next-best method's performance, i.e., that the next-best method is near-optimal. This suggests that harder tasks may be required in order to illustrate a clearer performance difference between the methods in more than 1 task. It would therefore significantly improve the paper if a performance comparison was performed on a task on which all methods are known to be far from optimal (e.g. a more difficult task, or a task for which the optimal performance is calculable and reported), in order to rule out the case that the existing evaluation is limited to "solved" tasks.
- It seems fairly odd that the final action selection strategy used during planning is an exponentially weighted average of the initial actions (Alg. 2, line 19). I am aware that prior work has considered this strategy (e.g. Charlesworth & Montana 2020), however, the MuZero MCTS used an argmax over an upper-confidence bound (Eq. 2 from https://arxiv.org/pdf/1911.08265.pdf). The paper would be improved with an experiment that varies the action selection strategy between the current one, the originally-proposed one from MuZero, and perhaps also a sampling-based one, drawing $a$ from the categorical distribution defined with unnormalized mass per bin $c$ as $\kappa \cdot R_c$.  Furthermore, it would improve the paper to run the method on a non-continuous control task, because I suspect that tasks with discrete actions would pose more of a challenge to the action-blending strategy used here.
- The employed dynamics models and environments are deterministic (aside the experiment in 5.3, in which the environment is deterministic), which limits our understanding of how well the approach would work in more complex environments with significant stochasticity.
- Eq. 2 is not self-contained: $l^p$, $l^v$, and $l^r$ are not defined in this paper. The paper should be self-contained.
- Missing citations to MCTS on p3: [A]
- p3-p4: The paper is not clear which rewards are used to compute the cumulative return $R(a_t^c)$. It seems that it could be either rewards computed using the original sparse reward function, or rewards computed using the reward model. I found the answer (the former) by looking at the supplement. The main text should be updated to clarify this uncertainty.
- Fig. 3 says there are "six scenarios" for reanalysis, but it's not clear that the six scenarios presented in Fig. 3 actually cover the space of possible outcomes of reanalysis. Specifically, the scenario in which the trajectory plans into an obstacle and the uncertainty ranges are small is not mentioned (i.e. the uncertainty estimate is inaccurate) -- instead, only the scenario in which the trajectory plans into an obstacle and the uncertainty range is large is presented (i.e. the uncertainty estimate is accurate). Either the caption should be fixed or those additional scenarios (one for intra-trajectory and 1 for inter-trajectory) should be added to the figure. Perhaps one way to add it would be to add other trajectories to a-2 and b-2 that intersect the obstacle, do not achieve the goal, and have low uncertainty.

## Citations
- [A] Coulom, Rémi. "Efficient selectivity and backup operators in Monte-Carlo tree search." International conference on computers and games. Berlin, Heidelberg: Springer Berlin Heidelberg, 2006.

---

> ### Author Response · Authors · 2024-04-03
>
> We sincerely appreciate the time and effort you dedicated to reviewing our paper. We have addressed each of these points individually and revised the paper in blue accordingly. We hope that our responses will effectively resolve your concerns.
>
> **W1** We agree that there is not substantial room for improvement in the current offline GCRL benchmarks (normal dataset results in Table 1), since the dataset is large and collected by an expert behaviour policy. Creating a more challenging benchmark for offline GCRL would be an interesting work for the current community. We have been aware of this point, thus we designed other benchmarks to evaluate our method. For example, GOPlan shows clear advantages in challenging tasks with small dataset (FetchSlide-s, HandReach-s); it also demonstrates its ability to against environment stochasticity (Figure 5); further, it has competitive performance to reach goals that do not appear in the dataset (Figure 4 in the main paper and Table 5 in Appendix). To further solve the concern, we design tasks with extra small datasets, whose size is 1/100 of the normal one, i.e., 20,000 transitions (or 400 trajectories). The results are shown in Table: Results on extra small datasets. GOPlan shows significant improvement compared to those (weighted) imitation methods, such as GCSL, WGCSL.
>
> | Strategy | GOPlan | BC | GCSL | WGCSL | GEAW | AM | CRL | g-TD3-BC | g-TT |
> |----------|--------|----|------|-------|------|----|-----|----------|------|
> | FetchPush-es | **18.91±1.2** | 10.57±1.9 | 5.96±2.1 | 13.25±2.8 | 10.35±2.3 | 11.28±1.6 | 16.13±1.8 | 6.27±0.9 | 15.76±1.2 |
> | FetchPick-es | **14.19±0.9** | 4.16±2.0 | 1.67±0.8 | 6.49±1.7 | 3.17±0.3 | 4.16±1.3 | 13.12±0.8 | 5.96±2.2 | 12.28±1.4 |
>
> *Table: Results on extra small datasets. The standard deviation ±0 means its value is less than 0.05.*
>
> **W2** In continuous control tasks, previous methods prefer to use the exponentially weighted average of the initial actions (Zhan et al., 2022; Charlesworth and Montana, 2020; Nagabandi et al., 2020), since it tends to avoid extreme value and produce a smooth action. We made further experiments to show GOPlan with different action selection strategies in reanalysis stage: select the action with the maximum score (`max_action`), sample the action with different weights (`sampled_action`), and draw the action from a categorical distribution (`categorical_action`). In particular, the `sampled_action` is sampled from the initial actions according to the weight $e^{kR(a^c_t)}/\sum^C_{c=1} e^\kappa$, where $R(a^c_t)$ is the normalized score for the initial action $a^c_t$; For the `categorical_action`, we partition every dimension of the action space into 10 bins, select the bin according to the average score of the initial actions falling into the bin, and take the centre of the bin as the `categorical_action`. The results are shown in Table: Model-based planning with different action selection strategies. `weighted_action` and `max_action` show similar good performance on both environments, while `sampled_action` is slightly worse in FetchPick and `categorical_action` is the worst strategy.
>
> | Strategy | FetchPush | FetchPick |
> |----------|-----------|-----------|
> | weighted_action | 39.15±0.6 | 37.01±1.1 |
> | max_action | 39.36±0.5 | 36.80±2.2 |
> | sampled_action | 39.25±1.2 | 35.45±2.7 |
> | categorical_action | 33.43±2.0 | 27.75±3.0 |
>
> *Table: Model-based planning with different action selection strategies.*
>
> **W3** We have been aware of the importance of an algorithm against stochasticity. In Section 5.3, we evaluate GOPlan in a stochastic FetchReach environment, in which different levels of Gaussian action noise is applied to the policy action outputs (0.5, 1.0 and 1.5). The offline dataset is also collected from a stochastic environments. The result shows GOPlan outperforms GCSL, WGCSL and AM. Besides, GOPlan keeps a moderate performance under a noise level of 1.5.
>
> **W4** In continuous control settings, the loss functions $l^p(\cdot, \cdot)$, $l^v(\cdot, \cdot)$, $l^r(\cdot, \cdot)$ are mean square error (MSE) between the inputs. We have corrected this in the revised version.
>
> **W5** Thanks for pointing out the related work. The suggested paper (Coulom, 2007) combines tree search with Monte-Carlo evaluation, which is an alternative model-based planning method for our finetuning stage. We have cited this paper in the revision.

---

> > ### Author Response · Authors · 2024-04-03
> > **Official Comment by Authors (cont'd)**
> >
> > **W6** The cumulative return $R(a^c_t)$ is computed by the sparse reward function. In goal-conditioned RL, this reward function is accessible to the agent, so we don't need a reward model to predict the reward. We have clarified this in the model-based planning part and the preliminary part.
> >
> > **W7** To the best of our knowledge, we have introduced all scenarios that are necessary in the reanalysis stage. The comment suggests that scenario where the trajectory plans into an obstacle and the uncertainty is small. However, it is not possible to have the scenario, given an accurately-estimated uncertainty. Since no trajectory in the offline dataset consists of states in the obstacles, the disagreement of the predictions of the dynamics models could be large in the states, resulting in a large uncertainty (Kidambi et al., 2020; Zhan et al., 2022). That means states in an obstacle *result in* large uncertainty.
> >
> > ---
> >
> > **References**
> >
> > Charlesworth, H., and Montana, G. (2020). PlanGAN: Model-based Planning with Sparse Rewards and Multiple Goals. In *Advances in Neural Information Processing Systems*.
> >
> > Coulom, R. (2007). Efficient Selectivity and Backup Operators in Monte-Carlo Tree Search. *Computers and Games*. Springer Berlin Heidelberg.
> >
> > Nagabandi, A., Konolige, K., Levine, S., and Kumar, V. (2020). Deep Dynamics Models for Learning Dexterous Manipulation. In *Conference on Robot Learning*.
> >
> > Kidambi, R., Rajeswaran, A., Netrapalli, P., and Joachims, T. (2020). MOReL: Model-Based Offline Reinforcement Learning. In *Advances in Neural Information Processing Systems*.
> >
> > Zhan, X., Zhu, X., and Xu, H. (2022). Model-Based Offline Planning with Trajectory Pruning. In *International Joint Conference on Artificial Intelligence*.

---

### Decision · Action_Editor_hmns · 2024-05-01

**Recommendation:** Accept with minor revision

**Comment:**

The paper offers a new methodology for learning and refining offline goal-conditioned RL policies -- leveraging planning techniques over a learned dynamics model and initial policy to generate additional synthetic trajectories. Results suggest the method is sound and empirically comparable to state-of-the-art baselines. The AE recommend a minor revision to ensure the additional experiments provided in the author response are fully incorporated into the manuscript prior to acceptance.

**Audience:**

Reviewers unanimously agree that there is an audience at TMLR for this work and the AE agrees. The proposed offline goal-conditioned RL framework explores an alternative research direction that seems comparable to previous methods in terms of raw performance and potentially provides some desirable properties for out-of-distribution goals.

**Claims And Evidence:**

Reviewers unanimously agree that the main claims are well-supported by the empirical results in the manuscript. The additional results provided in the response to reviewer 5fHZ provide further support that are welcome additions to the manuscript.